# Modified Nucleic Acids: Expanding the Capabilities of Functional Oligonucleotides

**DOI:** 10.3390/molecules25204659

**Published:** 2020-10-13

**Authors:** Steven Ochoa, Valeria T. Milam

**Affiliations:** 1School of Materials Science and Engineering, Georgia Institute of Technology, Atlanta, GA 30332, USA; sochoa8@gatech.edu; 2Petit Institute for Bioengineering and Bioscience, Georgia Institute of Technology, Atlanta, GA 30332, USA

**Keywords:** antisense oligonucleotides, aptamers, click chemistry, DNAzymes, polymerase, SELEX, XNA, XNAzymes

## Abstract

In the last three decades, oligonucleotides have been extensively investigated as probes, molecular ligands and even catalysts within therapeutic and diagnostic applications. The narrow chemical repertoire of natural nucleic acids, however, imposes restrictions on the functional scope of oligonucleotides. Initial efforts to overcome this deficiency in chemical diversity included conservative modifications to the sugar-phosphate backbone or the pendant base groups and resulted in enhanced in vivo performance. More importantly, later work involving other modifications led to the realization of new functional characteristics beyond initial intended therapeutic and diagnostic prospects. These results have inspired the exploration of increasingly exotic chemistries highly divergent from the canonical nucleic acid chemical structure that possess unnatural physiochemical properties. In this review, the authors highlight recent developments in modified oligonucleotides and the thrust towards designing novel nucleic acid-based ligands and catalysts with specifically engineered functions inaccessible to natural oligonucleotides.

## 1. General Introduction

DNA serves as the cellular vault where coded instructions for cellular activity can be safely stored and accessed. This code, however, does not require broad diversity in either its chemical makeup or its macromolecular structure for cellular enzymes to access and convert prescribed portions of this alphabetical code into protein production essential to normal cell housekeeping activities. While DNA and RNA share similar four-letter alphabets and Watson–Crick base-pair matching events to read, transcribe and translate this code, nature has assigned RNA with additional cellular tasks that require more functional abilities. By folding into more complex, diverse structure conformations than DNA, particular RNA structure-function motifs emerge to enable, for example, the specific protein binding events in a ribonucleoprotein or the catalytic capabilities of a ribozyme. This rich functional diversity of RNA supports its biologically important roles within gene expression, protein synthesis and intracellular communication. Collectively, the multiple roles of natural oligonucleotides inherent to a healthy host provoked researchers to begin deliberately harnessing the remarkable physiochemical properties of nucleic acids as biomacromolecular therapeutics.

The earliest therapeutic oligonucleotides, termed antisense oligonucleotides (ASO), primarily relied upon following complementary base pairing rules to regulate mRNA and subsequent protein processing implicated in disease progression. The first efforts employing ASO inside living cells date back to 1978 [1]. Outside of their protective cellular organelles, however, DNA and especially RNA are susceptible to chemical degradation by enzymes called nucleases or to rapid filtration out of the bloodstream by renal clearance. Consequently, naked oligonucleotides are rendered impractical for clinical use. Valued for their promise as therapeutic agents, researchers were, nevertheless, motivated to enhance the chemical resilience of ASO sequences in vivo while still relying on their natural base recognition capabilities to form duplexes with their complementary mRNA target to regulate its downstream cellular activity. For this historical reason, modifications to nucleic acids were initially directed towards increasing their resilience to nuclease cleavage to extend the half-life of injected ASO during their transport and circulation. With this singular goal, these efforts often focused on minimal, but effective, chemical alterations to the two ends of a sequence where many nucleases typically first bind. The most popular of these end modifications involved 3′ end capping with an inverted thymidine or chemical conjugation of larger macromolecules, especially polyethylene glycol (PEG), at the 5′ end to prevent renal clearance [2].

The emergence of solid-phase synthesis (SPS) coupled with benchtop polymerase chain reaction (PCR) techniques allowed for the large-scale production of nucleic acids by the late 1980s. A decade later the foundational work of three separate teams led to the realization that nongenomic DNA and RNA sequences could be selected as binding agents or ligands for other biological, non-nucleotide targets using an in vitro evolution-inspired screening platform [3,4,5]. Starting from large libraries of sequence candidates, this iterative in vitro screening process yielded novel ligands called aptamers for a wide variety of biological and later nonbiological targets. The in vivo performance of therapeutic aptamers composed of natural nucleotides, however, also suffered from rapid nuclease degradation. Similar in vitro screening procedures and in vivo performance outcomes followed suit for DNA sequences called DNAzymes which bind to a non-nucleotide target to catalyze a reaction. Thus, many of the lessons learned from modifying ASO sequences for in vivo stability were promptly applied to the sugar-phosphate backbones of aptamers and DNAzymes. These modification approaches were generally conservative to minimize unintended interference with target-binding or catalytic function of these functional oligonucleotides. Yet, it was later realized that in addition to attributing greater nuclease resistance, chemical modification could alter the intended properties of these functional oligonucleotides in an advantageous or disadvantageous manner (e.g., greater vs. weaker target binding affinity; more or less efficient catalysis behavior). In addition, novel functionality in oligonucleotide sequences could be explored and possibly rationally designed by expanding the toolbox of chemical modification techniques to include completely foreign nucleotide building blocks. This aspiring goal has inspired researchers to undertake more ambitious modification approaches to design highly functional modified oligonucleotides that exhibit physiochemical properties unattainable by nature.

Though natural nucleic acids have been well-studied in their purely biological context and still serve as effective biomacromolecular templates, modified oligonucleotides offer distinct opportunities as well as challenges to implementation as synthetic biomolecular tools. Rather than presenting an exhaustive review spanning the last several decades of progress in nucleic acid chemistry, the authors here instead aim to provide some historical context followed by a selective overview of recent interdisciplinary contributions from various science, technology and engineering fields to develop and study modified oligonucleotides. Notably, while multiple well-cited or historic references as well as review articles are included to provide historic context, technologically mature areas such as commercially available chemical modifications at the 3′ and 5′ ends (e.g., fluorescent dyes, biotinylated, etc.) have been thoroughly covered previously and thus are not covered here [6].

### 1.1. Overview of Historically Popular Backbone Chemical Modifications Intended to Enhance In Vivo Performance of Modified Oligonucleotides Acting as ASO or Aptamer Sequences

Owing to their non-immunogenicity, low toxicity and reproducibility, oligonucleotides historically serving as an ASO (via duplex formation with targeted RNA sequence) and as an aptamer (via single-stranded, but self-hybridized sequence binding to non-nucleotide target) have been intensely studied as capture or affinity reagents in bioanalytical and therapeutic applications. Since ASO sequences and aptamers derived from natural nucleic acids perform poorly in vivo, early modification strategies focused on increasing oligonucleotide stability by altering composition and structural features important to nuclease active site recognition, primarily along the sugar-phosphate backbone. Since efforts developing ASO preceded that of aptamers, many of the lessons learned from chemically modifying ASO sequences to confer their in vivo stability were promptly applied to the first aptamers intended as therapeutic agents. Generally, however, these modifications were solely purposed to support the intended aptamer-target binding function within biological environments rather than to explore or impart novel or additional functionality on candidate aptamer sequences.

Sequence parameters for any ASO are straight-forward since the ASO acts as a capture agent or binding agent for another oligonucleotide, typically RNA. Thus, the candidate sequence or primary structure for an ASO is predictable—it is simply the complementary hybridization partner to the targeted RNA sequence. In contrast, without any a priori knowledge of a suitable oligonucleotide-based binding agent or ligand for a non-nucleotide target, an aptamer sequence must be identified by diligently screening through large libraries of random sequences. Pioneering successes in the aptamer field in 1990 by three separate teams has led to a three-decade old screening platform known as systematic evolution of ligands by exponential enrichment (SELEX) [3,4,5]. This evolutionary screening platform begins by incubating the desired target with a large population of diverse oligonucleotides containing ~10^9^ to 10^15^ heterogeneous DNA or RNA sequences. This starting random pool is then subjected to multiple selection rounds intended to separate functional, target-bound sequences from inactive, unbound sequences. Following each target incubation step in a given round, the target-bound sequences are intentionally dissociated or separated from the target, then amplified by PCR to enrich the candidate population pool with sequence winners from a given screening round. This enriched sequence population, in turn, generates the next iteration of promising aptamer candidates for subsequent screening round(s). After several repetitions of these tightly coupled, alternating steps of target incubation and PCR enrichment, the once large, diverse, random pool of oligonucleotides ideally converges upon one or more sequences possessing the intended target recognition and binding capabilities. To date, countless DNA and RNA aptamers for a variety of biomolecular, protein and cellular targets have been reported in the literature using a SELEX-based screening platform. A similar SELEX platform is later employed to identify a separate, but related category of sequences called DNAzymes with specific catalytic capabilities in the presence of a target [3]. Further discussion of DNAzymes is provided in later subsections.

Aside from capping the 5′ or 3′ end, chemical stability-enhancing modification routes within the sequence itself were initially restricted to substituting select positions in the sugar group or phosphodiester bond as illustrated in Figure 1 and listed in Table 1 [2]. Other studies involved modifications of particular numerical positions in nucleobases (e.g., 2,6-diamino purines; C5 modification in pyrimidines) to yield more stable duplexes [7]. Further discussion of modifying these positions or even completely replacing nucleobases is provided in later subsections in this review. As discussed in the next two subsections, while desirable to inhibit the binding and cleavage activity of nucleases to oligonucleotides, the historic popularity associated with modifying specific positions in either the sugar group or phosphodiester bond with particular chemical groups stems from the need for chemically modified sequences to avoid nuclease binding, yet still enable binding and activity of key enzymes called polymerases used to amplify the copy number of an oligonucleotide sequence. As a processing tool for numerically enriching sequence representation within a large heterogeneous sequence population, polymerases play an essential role as both (1) a prerequisite reagent for preparing the initial random sequence screening libraries as well as (2) an intermittent reagent between each selection round during SELEX screening [3,4,5]. On the other hand, post-SELEX modification of RNA or DNA aptamers can, to some degree, sidestep potential sequence-polymerase compatibility issues. Thus, as illustrated in several examples involving modified sugar groups discussed next, polymerase incompatibility is a crucial barrier to intermittent PCR steps inherent to SELEX platform if the screening library itself is comprised of modified oligonucleotides.

### 1.2. Chemical Substitutions and Bridges in Pentose Sugar Groups

The sugar moiety is a frequently modified component of the tripartite nucleic acid structure and represents a straightforward method for promoting in vivo stability. As illustrated in Figure 1, sequences can be modified by substituting the 2′O position of the ribofuranose ring with fluoro (-F), amino (-NH_2_), azido (-N_3_) or methoxy/OMe (-OCH_3_) groups [28] with different effects on hydrogen bonding properties and resulting sugar pucker conformation which, in turn, affect the stability of the helical structure [17]. As indicated in Table 1 though nuclease resistance is enhanced in all these modification approaches, the thermal stability effects can vary widely with 2′-NH_2_ groups reportedly destabilizing helical structures and 2′-F and 2′-OMe groups reportedly enhancing helical structures [17]. Polymerases capable of accommodating 2′-F and 2′-OMe nucleotides have been used to prepare aptamer screening libraries comprised of chemically modified, randomized sequences [9,10,18]. In particular, 2′-F sugar modifications have been utilized for over two decades due to their favored SPS efficiencies as well as their high degree of compatibility with polymerases. In fact, the only FDA approved aptamer, Pegaptanib, is a 27 nucleotide long RNA sequence that employs a combination of 2′-F pyrimidines, 2′-OMe purines and an inverted 3′ end thymidine to maximize in vivo performance as an antagonist to vascular endothelial growth factor (VEGF) [29]. It is important to note that incorporation of modifications within the binding motifs of aptamer sequences can disrupt necessary conformations resulting in a loss of aptamer affinity. For this reason, aptamer sequences modified post-SELEX require systematic substitution of modified nucleotides at specific positions followed by investigation of any effects on binding affinity. Consequently, in earlier work, rather than incorporating all the different chemical modifications in the screening library itself, some chemical modifications (e.g., 2′-OMe substitutions at particular nucleotide positions) used in the Pegaptanib aptamer were introduced post-SELEX into an aptamer selected via in vitro SELEX [17]. Mi et al. conducted the first in vivo SELEX-based selection experiments with modified oligonucleotides by substituting 2′-F pyrimidines for natural pyrimidine nucleotides in RNA sequences to successfully screen against an enzyme upregulated in cancer tissue in tumor bearing mice [30]. Mi et al. later refined their in vivo SELEX platform to include implanted xenografts from human cancer patients into mice hosts [31].

Instead of introducing a substitutional chemical group at one specific sugar ring position, others sought to structurally limit conformational freedom by introducing a covalent bridge across two specific positions in the pentose ring structure. For example, as illustrated in Figure 1, a methylene bridge between the 2′O and 4′C of the sugar ring conformationally locks the sugar into an N-type pucker [11], resulting in a locked nucleic acid (LNA). In contrast to the varying thermal stability effects of specific chemical substitutions in 2′O position of sugar groups discussed above, LNA substitutions result in oligonucleotides with both higher duplex stability and superior nuclease resistance [32,33,34]. Despite differences in their duplex densities, high throughput flow cytometry studies of microspheres functionalized with single-stranded DNA or DNA/LNA mixmer probes indicated that the kinetics of duplex formation with either pure DNA or DNA/LNA mixmer targets were comparable across nearly all sequence combinations [35]. In contrast to the more extensive work with 2′-F sugar modifications in aptamers, less work has been reported with aptamers possessing LNA though work by Veedu et al. indicates commercial polymerases used for PCR can tolerate several LNA substitutions in DNA [36]. One exception to this overall scarcity in reported LNA aptamers, however, is work by Shi et al. who modified a DNA aptamer called TD05 post-SELEX for lymphoma Ramos cell targets by substituting 14 nucleotide positions with LNA in addition to including an inverted 3′ thymidine end cap. Their modified TD05 derivative exhibited a serum half-life up to ten times greater than the natural cognate [37]. Despite its overall scarcer implementation to date, the relative ease of handling LNA substitutions coupled with their advantageous physicochemical properties merits further exploration of LNA in various nucleic acid systems including ASO and aptamers.

### 1.3. Phosphodiester Linkage Modifications and Complete Backbone Replacement Strategies

Another common strategy used to augment oligonucleotide in vivo stability relies on chemically altering the phosphodiester linkage between nucleotides. Nucleic acid analogs with a phosphorothioate (PS) backbone [22] illustrated in Figure 1 as well as a methylphosphonate backbone [38] can be created by replacing a non-bridging oxygen atom in the phosphate backbone with a sulfur atom or uncharged methyl group, respectively. Of the multiple replacement possibilities shown in Figure 1 for the phosphodiester bond, PS sequences are historically the most popular. Synthetic oligonucleotides called thioaptamers which incorporate this sulfur atom substitution have been shown to enhance in vivo stability although they can suffer from reduced non-nucleotide target binding specificity if simply modifying the original DNA aptamer (i.e., post-SELEX) with its PS analogue [39]. Intriguingly, this same study by Wu et al. indicated that post-SELEX modification of the same original DNA aptamer with 2′-OMe groups on sugar groups did not reduce target binding affinity. Similar to libraries with 2′-F pyrimidines for in vivo screening sessions, single-stranded libraries of PS candidates have also been employed for in vivo screening [40].

In contrast to modifying either the sugar or phosphate group, completely replacing both the sugar and phosphate groups with a peptide backbone while retaining natural nucleobases provides peptide nucleic acids (PNA) with backbones unrecognizable by nucleases. Lee et al. synthesized a 15-mer PNA strand with an otherwise identical nucleobase sequence as the well-known thrombin binding aptamer, TBA15 and reported comparable performance in its selective binding activity to thrombin [41]. In contrast to employing a peptide backbone, Varizhuk et al. replaced the phosphate backbone of TBA15 with a triazole inter-nucleotide linkage shown in Figure 1 and similarly found that the resulting synthetic polymer was able to bind thrombin, but with increased resistance to nuclease hydrolysis [14].

In many of the above scenarios, chemical modifications have structural implications that affect its behavior. For example, though often pictorially depicted as a planar molecule, the sugar group in unhybridized nucleotides has conformational freedom to explore different puckering conformations known as S (for South) and N (for North). This conformational freedom is lost upon hybridization with its Watson–Crick pair. As mentioned previously, the inclusion of a chemical bridge across 2′O and 4′C permanently locks the sugar group in LNA into an N conformation in both its hybridized (i.e., paired with its complementary base) and unhybridized state. For other backbone modifications, additional structural implications can arise. For example, replacing one oxygen atom with a sulfur atom renders the P atom as a chiral center at each modified phosphate group. Thus, numerous diastereomers can result, even in a short oligonucleotide in which each phosphate group is replaced by phosphorothioate. This complication is problematic, particularly since stereoselective PS synthesis is challenging and can affect its therapeutic efficacy [42,43]. Recently, however, reagents enabling a relatively simple synthesis of enantiomerically pure methylphosphonate oligonucleotide precursors have been reported as a first key step towards synthesis of these challenging chiral nucleotides [44,45]. The next subsection discusses an intentional modification approach that stems strictly from changing the spatial placement of atoms (rather than their chemical identity) in an oligonucleotide.

### 1.4. Structural Modification Involving Mirror Image Analogs to Natural d-Oligonucleotides

A subclass of modified oligonucleotides known as Spiegelmer^®^ sequences, exemplify the ideal rationale behind chemically modifying aptamer sequences as a means to enhance in vivo stability without compromising its target-binding function. Spiegelmer^®^ sequences are the chiral enantiomers (i.e., unnatural L-oligonucleotide) of a natural oligonucleotide sequence (i.e., d-oligonucleotide) that exhibit extended half-lives in vivo because they are unrecognizable by nucleases. These mirror-image oligonucleotides have been successfully commercialized by NOXXON Pharma with two mirror-image aptamers (targeting several key chemokine signaling proteins) awaiting FDA approval. Chiral molecules such as individual amino acids [46], polypeptides [47], RNA structures [48] or select molecular metabolites [49] are the most suitable target choices. To facilitate initial SELEX screening, ideally one uses a library comprised of the natural oligonucleotides to identify DNA or RNA ligands for the unnatural target enantiomer. After selecting a suitable DNA or RNA aptamer, the sequence is then converted into its Spiegelmer^®^ analog to promote binding to its natural target enantiomer. This screening and sequence conversion approach requires complete knowledge of the target structure and thus limits the target spectrum available to Spiegelmers [50]; however, in 2016 Wang et al. [51] followed by others [52,53] reported successful recognition and transcription of L-oligonucleotides by a novel synthetic polymerase (comprised of unnatural d-amino acids) which could widen the applicability of Spiegelmer^®^ sequences.

### 1.5. Performance of Modified Oligonucleotides as Primary Hybridization Partners for Antisense Therapeutics and Gene Editing

As of early 2020, the vast majority of the eleven FDA-approved modified oligonucleotides serve as antisense oligonucleotides (ASO) in which the oligonucleotide therapeutic is intended to inhibit activity of messenger RNA (mRNA) by forming an ASO:mRNA duplex [54]. By inhibiting the normal role of mRNA during translation in ribosomes, the ASO can thus influence protein expression that often serves as the source of disease pathogenesis. To serve as a chemically durable hybridization partner that survives initial transport through the extracellular environment, the ASO must exhibit nuclease resistance and sufficient or, preferably enhanced binding affinity for its RNA target once reached. To achieve these robust biocompatible properties, several of these eleven FDA-approved oligonucleotide therapeutics rely on incorporating the same modification such as a continuous PS backbone throughout the sequence (e.g., Milasen and Formivirsen (Virtravene)) while other oligonucleotide sequences are comprised of different, position-dependent modifications (e.g., Mipomersen (Kynamro) possesses a central DNA segment flanked by five modified sugars on each end). On the other hand, PS backbones tend to weaken the stability of the resulting PS:target duplex by lowering its melting temperature [55]. Despite this consequence, PS backbones remain among the most popular chemical modification approaches for ASO sequences.

Rather than inhibiting mRNA activity after DNA encoding information has already been accessed, a more recent biotechnological application employs modified oligonucleotides to edit the genetic code itself in nuclear DNA [56]. Similar to traditional ASO sequences targeting mRNA outside the nucleus, the advantage to using modified oligonucleotides as a direct gene editing tool stems first from their nuclease resilience during an even more complicated passage to gain nuclear entry [57]. While higher target affinity is often intrinsic to many chemically modified oligonucleotides, the variability in off-target effects during gene-editing steps is still a challenging aspect [58] requiring further testing of singular or combinations of chemical modifications in sugar and/or phosphodiester groups in the same sequence [23,57]. Notably, in contrast to traditional ASO targeting of transient mRNA, successful gene edits are permanently encoded into the cell and thus automatically inherited by daughter cells during cell division.

While the above studies rely on hybridization events in order to regulate subsequent cell housekeeping activities, other studies focus on enabling simple, yet effective in vitro cellular or acellular, single-stranded probes or capture agents that can bind specific natural oligonucleotide targets as illustrated in Figure 2. Modified oligonucleotides such as LNA have demonstrated their abilities as thermally robust capture agents of natural oligonucleotides present at low copy numbers [59]. Comparison of probes incorporating different modified nucleotides also indicates better mismatch discrimination capabilities for LNA, 2′-OMe and PNA over pure DNA probes; however, these same studies indicate that probes with unlocked nucleic acids (UNA) perform more poorly as capture agents for mismatched targets [60]. These thermodynamic studies suggest UNA allows for hydrogen bonding arising from Watson–Crick base-pair matching; however, in contrast to the locked conformation of the cyclic sugar group in LNA, the intrinsic flexibility of the acyclic sugar group in UNA could disrupt pi-pi stacking of base-pairs to cause duplex destabilization.

### 1.6. Performance of Modified Oligonucleotides in Double-Stranded Probe Systems for Displacement Strategies

In addition to serving as the sole hybridization partner for a natural oligonucleotide, modified oligonucleotides can also serve as a temporary primary hybridization partner susceptible to exchange with a higher affinity (e.g., more base pair matches) secondary or competitive hybridization partner. In these displacement scenarios illustrated in Figure 2, a single-stranded segment called a toehold domain in the original duplex can serve as a nucleation site for the competitive hybridization partner to initiate secondary duplex formation. For modified oligonucleotides such as LNA, their enhanced binding affinity for a given sequence allows for a shorter, yet thermally stable LNA sequence to serve as the temporary primary hybridization partner. This allows for a longer, single-stranded toehold segment to be incorporated in the original or primary duplex and enable effective nucleation and displacement by a longer competitive natural oligonucleotide such as a DNA fragment from chromatin [19]. In fact, with a sufficient toehold base length, DNA/LNA mixmers (instead of pure LNA sequences) can serve as effective primary or competitive secondary targets. Using a large combinatorial array of different toehold lengths and position-dependent LNA substitutions, Olson et al. demonstrated the range of displacement kinetics possible in oligonucleotide solutions [62]. By immobilizing nearly complementary DNA/LNA mixmers on microspheres and nanoparticles, Eze and Milam formed colloidal satellite assemblies susceptible to displacement-driven colloidal disassembly by perfectly complementary DNA/LNA targets under isothermal conditions [34]. Using chimeric blends of right-handed DNA and left-handed DNA in their double-stranded probes, Young and Sczepanski added an elegant chirality-dependence to their displacement approach [63]. Notably, in contrast to achiral PNA sequences introduced in the previous section, the toeholds and hybridization segments within these chimeric blends will only hybridize to a complementary sequence segment of the same chirality.

### 1.7. Current Limitations of Popular Chemical Substitutions and Preview of Recent Approaches Involving Artificial Chemical Groups

Decades of studies have refined synthetic oligonucleotides intended for therapeutic applications as ASO and aptamers. The initial goal was to ensure minimal nuclease activity with sequences and then ensure sufficient target affinity. For ASO applications, these studies narrowly focused on defining nucleic acid functionality by its endpoint ability to bind to the oligonucleotide target (e.g., mRNA) itself; however, a series of recent publications by Crooke and colleagues [64,65,66] examines the importance of specifically understanding the role of ASO interactions with other proteins encountered on its pathway to the desired oligonucleotide target. As just one example, in addition to rendering the P atom as a chiral center in the phosphate group, the sulfur atom substitution in PS reportedly “spreads” the negative charge and makes the phosphate group itself more lipophilic [67]. Thus, while resistant to nuclease binding and cleavage activity, this enhanced lipophilicity tends to actually promote protein binding to a PS sequence intended to function as an ASO. Given the rich diversity of extracellular and intracellular proteins, there is no singular or universal ASO-protein interaction though general principles have emerged regarding cellular uptake and trafficking of ASO sequences alone as well as part of a co-delivery or carrier system [68].

Compared to the number of studies detailing effects on oligonucleotide properties such as nuclease resistance and melting temperature of duplexes comprised of complementary unnatural and natural oligonucleotides as summarized in Table 1, there are far fewer reports [69,70,71] specifically addressing immunostimulatory effects of isolated or specific chemical modifications. Most of the nucleic acid sequences discussed in this manuscript are only tens of bases long and thus too small in molecular weight to induce a significant immune response. One exception, however, involves larger 3D polyhedral frameworks called nucleic acid nanoparticles (NANP). Since most NANP in the literature are comprised of natural oligonucleotides [72], they are not discussed in detail here, but their immunostimulatory properties have been documented in other reviews [73].

For aptamers, while the need for in vivo stability also motivated initial chemical modification efforts, modifications under this regime also investigated a relatively narrow set of chemical parameters to support its target-binding function within biological environments rather than to impart additional or novel functionality. Furthermore, the scope of alterations to the nucleic acid structure in this context was generally restricted to the sugar-phosphate backbone because of its significance in minimizing nuclease active site recognition. As mentioned previously, another important hurdle relevant to aptamer screening arose from the prerequisite role of polymerases. The high fidelity of natural polymerases in amplifying the copy number of a particular sequence is a desirable property, especially in processes such as SELEX which rely on PCR-based enrichment of aptamer candidates during screening cycles as well as actual post-SELEX sequencing of DNA and RNA strands recovered from the screening process. This same fidelity, however, also means that polymerases do not readily recognize or incorporate highly divergent nucleotide chemistries which, in turn, limits the chemical diversity of screening libraries for in vitro and in vivo selection methods [74].

Furthermore, while highly modified oligonucleotides can be achieved via SPS post-SELEX, these subsequent chemical modifications to the original DNA or RNA aptamer can compromise the intended target-binding activity of the oligonucleotide [75,76,77]. Despite this limitation in overall library diversity, it should be noted that numerous aptamers generated from modified oligonucleotide libraries (rather than post-SELEX) have been reported as summarized in a comprehensive 2016 article by Lipi et al. [78]. While the need for in vivo stability thus drove initial efforts to modify nucleic acids without deviating from the canonical structure of nucleic acids, the next subsections focus on later studies which explored bolder chemical modifications to the nucleic acid scaffold ranging from simpler “click” chemistry using alkyne handles to add groups to otherwise natural nucleobases to incorporating completely artificial nucleobases within a sequence. With this richer chemical diversity, these bio-inspired macromolecules could be exploited to enhance not only their target-binding capabilities, but potentially unmask additional advantageous functionalities of nucleic acids for existing and emerging diagnostic, sensing and bioanalytical applications.

## 2. More Recent Efforts to Expand Oligonucleotide Modification Approaches to Enhance Their Functionality

Though natural oligonucleotides are limited to four canonical nucleobases, their functionality in natural biological systems has evolved over a billion years from short, yet functional RNA to much longer DNA sequences with larger storage capacity. Evolutionary aspects on a much shorter time scale (~weeks) are mimicked in a laboratory setting using SELEX screening to select functional oligonucleotides called aptamers from among ~10^9^ or more random sequences. However, in contrast to the numerous combinatorial possibilities with amino acid building blocks, the four-letter alphabet of nucleobases intrinsically restricts the chemical diversity of these random sequence libraries. To overcome this limitation in chemical diversity, researchers have developed several strategies for modifying natural nucleobases with functional moieties in an effort to expand the potential physiochemical properties of nucleic acids.

Particular chemical backbone modifications discussed in the previous subsections were limited to those tolerated by natural polymerases. As discussed in the next subsections, progress in directed evolution methods has established new engineered polymerases that can accommodate a larger variety of foreign or artificial groups in the sugar-phosphate backbone making the direct selection of modified aptamers possible [9]. To assist the exploration of bulkier chemical modifications that are not compatible with currently available engineered polymerases, a few key contributors have developed clever workarounds that still permit the in vitro selection using densely modified sequence libraries.

### 2.1. Artificial Sugar Groups in Xeno-Nucleic Acids (XNA)

Herdewijn and Marliere defined the term xeno-nucleic acids or XNA as nucleic acids possessing “chemical backbone motifs [that] would differ from deoxyribose and ribose” [79]. While some literature does broaden this definition to include, for example, single-atom substitutions to any portion of the tripartite nucleotide [18], this subsection focuses specifically on sugar chemistries that are highly divergent from the traditional ribofuranose or deoxyribofuranose ring and thus bear minimal resemblance to their natural counterparts. For example, in lieu of the furanose ring, researchers have incorporated sugar groups such as threose in TNA [80] and even six membered ring structures like hexitol in HNA [12] illustrated in Figure 1. Polymerase evolution experiments have provided the means to incorporate several of these XNAs such as cyclohexene in CeNA and TNA within in vitro evolutionary schemes [8]. Arangundy-Franklin et al. reported engineering a highly mutated derivative of Tgo polymerase to synthesize a full-length XNA with a P-methyl/ethyl-phosphonate backbone (phNA). The authors then used a “DNA display” strategy to identify an aptamer specific to streptavidin, illustrating the potential for functional aptamers lacking the polyanionic backbone. Importantly, this oligonucleotide presents the first synthetic nucleic acid with a completely charge neutral backbone derived by enzyme mediated screening methods. Moreover, this approach widens the range of aptamer targets to include previously refractory target candidates such as anionic molecules [23].

Notably, in contrast to the previously discussed LNA which can be tolerated by natural polymerases for several PCR cycles, this subsection focused on XNA groups that are poor substrates for natural polymerases, yet still enable Watson–Crick base-pair matching events to occur. Even fewer exceptions of XNA recognized by natural reverse transcriptases, however, have been reported [81,82]. To address this gap in compatibility between reverse transcriptases and XNA, Houlihan et al. described a strategy using emulsion droplets as compartments for individual reactants for in vitro screening to identify several improved or completely novel reverse transcriptases for separately converting several XNA chemistries into DNA [83]. Notably, where applicable, natural reverse transcriptases appear to be limited to only recognizing modified nucleobases found in organisms rather than xenobiotic nucleobases which are completely artificial [84]. The next subsection discusses other chemical modifications that can be explored though several of these oligonucleotides lack engineered enzymes which can recognize specific chemical modifications.

### 2.2. Modifying Nucleobases with Singular Hydrophobic Groups, Chemical Handles or Carbohydrates

Aptamers modified with hydrophobic groups have found a particularly useful niche in the area of biomarker detection and even emerged as a commercial contender in the field of proteomics. The aptamer-based diagnostic company, SomaLogic, has garnered considerable commercial success preparing libraries in which one of multiple hydrophobic groups (e.g., benzyl, tryptamino) is conjugated to the C5 position in dU/C nucleobases, a position well tolerated by developed mutagenic polymerases allowing for adequate PCR amplification. The resulting modified oligonucleotides, equipped with amino acid mimics, are screened via SELEX to identify slow off-rate aptamers called SOMAmers. SomaLogic has established multiple SOMAmer based proteome assays with over 3,000 different protein targets identified so far [85,86].

To further assist the exploration of bulkier chemical modifications in aptamer candidates lacking engineered polymerases, a few key contributors have developed clever workarounds that still permit in vitro SELEX selection using densely modified libraries. A conceptual workaround leveraged a 2-amino substitution in pyrimidines as a synthetic handle for conjugating molecules and even bulky groups via aldehyde reactions [87]. Similar to the approach by Bugaut et al., dUTPs modified at the C5 position with alkyne or carboxamide groups can serve as other synthetic handles [88].

Expanding on these alkyne handles, a useful scheme introduced by Mayers’ group [89] realized modified nucleic acids nicknamed “clickmers” as scaffolds to which almost any azide-bearing functional group can be conjugated to an alkyne-modified (at C5 site) dU via facile copper-catalyzed azide-alkyne cycloaddition (CuAAC) or “click chemistry” [90]. These modular nucleic acid templates can be easily adapted with large chemical moieties such as long carbohydrate chains and polycyclic compounds that are normally incompatible with polymerase-mediated evolution methods [90,91]. Though a glycan chemical modification, for example, is identical at every dU nucleotide, Krauss’s group postulates that composition diversity in the random glycosylated sequence library itself yields structural diversity in how glycan groups are clustered within self-folded glycosylated oligonucleotides to enable a tighter fit to their HIV antibody target [92,93]. While click chemistry facilitates straight-forward modification at every dU nucleotide in a sequence, in order to PCR-enrich winners from each SELEX screening round as illustrated in Figure 3, each round requires “transcription” from the alkyne-modified precursor sequences into the desired modified oligonucleotides followed by “reverse transcription” back to precursor sequences possessing pendant alkyne groups. Thus, while this approach can be easily adapted to incorporate any one of a wide variety of nucleobase chemistries, only one particular chemical group addition can be accommodated in a given screening library. To further diversify the number of chemical modifications incorporated into an oligonucleotide screening library and consequently increase the explorable sequence space, others have expanded beyond the four-letter alphabet available with canonical nucleobases as discussed next.

### 2.3. Xenobiotic Nucleobases as Artificial Base-Pair Matches

In contrast to briefly mentioned position-specific modifications in nucleobases which do not alter Watson–Crick base-pairing interactions (see examples in Table 1), libraries of random oligonucleotides that include completely foreign or unnatural nucleobases serve as richer genetic pools from which synthetic aptamers with enhanced functionality can be evolved. While several examples of position-specific nucleobase modifications which still allow Watson–Crick base pairings are highlighted in Figure 1, rather than broadly applying the term “xenobiotic” to potentially refer to any portion of the nucleotide [94,95] the term “xenobiotic nucleobases” here will refer to completely foreign nucleobases with examples illustrated in Figure 4. An artificially expanded genetic information system, comprised of four natural bases (A, C, G, T) and an additional two xenobiotic nucleobases intended as an artificial Watson–Crick pair (trivially called Z and P), was used by Sefah et al. to identify a modified aptamer specific to a line of breast cancer cells with a reported dissociation constant of 30 nM [96]. Using the same six nucleotide library, Zhang et al. added a negative selection step in a laboratory in vitro evolution (LIVE) experiment to identify eight modified aptamers with dissociation constants of 10–100 nM for liver cancer cells (HepG2) [97]. In both studies, aptamers emerging from libraries (following 200 cycles of PCR reported by Zhang et al.) which contain only the four canonical nucleotides exhibited much lower binding affinities for the cellular target, indicating the superior molecular performance of the modified aptamer sequences is directly dependent on an expanded genetic set. Intriguingly, the higher affinities in Sefah’s studies were attributed to a single occurrence of the synthetic Z and P nucleobases within the aptamer sequence. Moreover, while the average starting library member possessed 3 Z and 1.5 P moieties, the reduction to a single Z and P nucleobase during SELEX screening was attributed to amplification inefficiencies during PCR. In contrast to including two unnatural nucleotides in the screening library, Kimoto et al. intentionally excluded the unnatural “Px” nucleotide during their aptamer screening in order to allow the other unnatural, highly hydrophobic base “Ds” to remain “unhybridized” or unmatched and thus more available to bind to hydrophobic cavities in the targeted protein [98]. Unlike the completely randomized central segments in the AEGIS libraries prepared by Sefah and Zhang, the single unnatural nucleobases in oligonucleotides studied by Kimoto et al. had to be included at predetermined positions in the sequence libraries due to their incompatibility with conventional cloning and sequencing methods. Later work by Hirao’s group reported an updated PCR approach to enable randomized placement of their unnatural, hydrophobic nucleobase [99].

### 2.4. Further Expanding Library Diversity Using Multiple Modifications “Coded” into DNA Sequence

In lieu of enlarging the sequence space with additional synthetic nucleobases, other investigators rely on modifying natural nucleobases with additional functional groups, such as side-chains, that extend beyond the repertoire of position-specific single atom or small group modifications illustrated in Figure 1. Moreover, in contrast to the click chemistry approach in which only one unnatural chemical group can be included, multiple unique chemical groups can be included by mimicking the role of codons in the DNA template sequences. Chen et al. attached eight different sidechains to the C5 position of pyrimidines to create a pool of artificial trinucleotides as anticodons [100]. Using natural DNA sequences as template strands combined with a ligase-mediated polymerization method, they then “translated” natural oligonucleotides into a library of antisense partners comprised of modified and natural nucleotide building blocks. Selection experiments with this library isolated an aptamer that binds its protein target with high affinity (reported *K_d_* of 3 nM) in which position-dependent placement of the side chains was credited with playing a key role in target binding success. Continuing earlier work to create an array of artificial pentanucleotides as anticodons [101], this translation approach was also employed by Hili’s group to identify aptamers comprised of natural and modified nucleotides against thrombin as illustrated in Figure 5. In contrast to the crucial role G-quadruplex formation plays in DNA aptamer-thrombin binding for the historic Bock aptamer [102], Hili’s group postulated that the resulting stem-loop secondary structure of their chemically diverse aptamer, the first example of a thrombin aptamer lacking the G-quadruplex conformational motif supports the hypothesis that “diversity begets function.”

### 2.5. Various Modified Aptamers as Protein Activity Regulators

Countless aptamer sequences emerging from SELEX screening studies have been reported over the last 30 years since the term “aptamer” was first coined. Aside from its sequence identity and binding affinity and in contrast to an ASO sequence interfering with protein expression by simply binding to its complementary mRNA sequence, fewer studies delve into understanding the nature of aptamer-target binding, especially for protein targets which have multiple potential binding sites. From a therapeutic stance, however, the ability to not only bind, but actually modulate protein activity, remains a more challenging role for modified aptamers. Recent progress has illustrated modified aptamers as antagonists which inhibit protein activity, possibly through allosteric regulation. For example, rather than employing completely foreign nucleobases, Gasse et al. employed more classic position-specific nucleobase chemical modifications (highlighted in Figure 1) by substituting thymidine and adenosine residues with C5 modified chloro-uracil and N7 modified deaza-adenine analogues, respectively [104]. When subjected to evolutionary screening against a BACE1 target, an enzyme involved in Alzheimer’s disease, this doubly modified library became enriched with multiple aptamer candidates including an aptamer that binds BACE1 with relatively high affinity (equilibrium dissociation constant, *K_d_* = 12 nM). Interestingly, this modified aptamer reportedly inhibits BACE1 enzymatic activity at low aptamer concentrations but displays an activating effect at higher aptamer concentrations. Later that year, Tan et al. combined multiple nucleotide modification approaches to include nucleotides with a ferrocenyl group, a trifluoromethyl group and a Z:P base pair in their screening libraries. They identified an aptamer against integrin alpha 3 that effectively inhibited tumor cell adhesion and migration [105]. Rather than completely blocking integrin binding sites, however, the authors speculated that bound aptamers may preferentially stabilize particular integrin conformations that, in turn, reduce integrin binding activity to one of its natural ligands, namely an extracellular matrix protein called laminin. Instead of relying on thermodynamic predictions of RNA and DNA self-hybridized structures, further advances in achieving allosteric regulation of a protein target requires developing specific, composition-dependent predictive tools for modified oligonucleotides, both alone and in the presence of its particular target.

## 3. Modified Nucleic Acid Enzymes

The complex folding of short DNA or RNA sequences makes them capable as ligands. A subset of the ligands called DNAzymes (for DNA sequences) or ribozymes (for RNA sequences) can achieve conformations promoting activity as catalysts. While ribozymes are found in nature, DNAzymes are absent in nature and are thus selected through in vitro selection in the laboratory setting [3]. Early nucleic acid catalysts suffered from poor in vivo performance due to their general dependence on divalent metal cations in concentrations higher than typical biological environments. Slow reaction rates, low catalytic efficiencies and single-turnover characteristics additionally rendered many of the first synthetic oligonucleotides inferior to proteinogenic enzymes. These shortcomings can be generally attributed to the restricted chemical diversity of natural nucleic acids, emphasizing the need to augment the physiochemical properties of DNAzymes. Chemically modified oligonucleotide catalysts appear to benefit from greater functional diversity as will be covered in greater detail in the next subsection. Notably, several additional examples of modified nucleic acid catalysts are thoroughly discussed in a 2019 review by Hollenstein [95].

### 3.1. Modified Nucleic Acid Enzymes with Efficient and Novel Catalytic Activity

Early approaches to improve the poor performance of modified nucleic acid catalysts attempted to replicate the active sites of natural enzymes such as metal independent ribonucleases by modifying the catalytic cores of DNAzymes with amino acid-like moieties [106,107]. Similar to the two broad approaches for implementing aptamers comprised of unnatural chemical components, pre-SELEX screening with modified libraries versus post-SELEX modification strategies were applied to modified nucleic acid enzymes [95]. By incorporating enzyme-resembling chemical moieties such as imidazoles, cationic guanidines, cationic amines and cationic histamine groups, investigators have successfully engineered efficient RNA cleaving modified DNAzymes [108]. Recently, Wang et al. used three modified dNTPs and dGTPs to synthesize an RNA-cleaving, modified DNAzyme capable of multiple catalytic turnover numbers without any divalent metal cation present, marking the first report of a metal-independent DNAzyme [109]. The addition of unnatural functional groups during in vitro selection experiments also enables the identification of DNAzymes that catalyze reactions otherwise elusive to natural nucleic acid sequences. For example, Zhou et al. modified the C5 position of thymidines with either amino, carboxyl or hydroxyl groups to synthesize a modified sequence screening library. Selection experiments yielded a DNAzyme that can catalyze the hydrolysis of aliphatic amide bonds, a feat that was unsuccessful in prior attempts with sequences comprised of natural nucleotides [110].

Intriguingly, from nucleobase modifications, alterations to the sugar-phosphate backbone can also promote catalytic function. Taylor et al. substituted the sugar ring with four alternate chemical structures with the resulting arabino, 2′-FANA, CeNA and HNA sequences capable of catalyzing RNA ligase and endonuclease reactions [111]. Notably, Taylor et al. refer to their nucleic acid enzymes as XNAzymes and FANAzymes. Researchers modified various peroxidase-catalyzing oligonucleotides with 2′-OMe sugar substitutions that form G-quadruplexes and reported significantly enhanced thermal stability, hemin-binding affinity and enzymatic activity [112]. An alternative approach to modified nucleic acid catalysts aims to overcome the chemical limitations imposed by the natural electronegative phosphate backbone, specifically in its weaker binding activity with thiophilic metal ions. For example, Huang et al. modified a DNAzyme by introducing a single PS modification along its backbone to enhance its affinity for Cd^2+^, a thiophilic metal [113]. The same group also reported a DNAzyme comprised of natural nucleotides and capable of cleaving a PS modified oligonucleotide [114].

### 3.2. Spatiotemporally Controlled Synthetic Catalysts

A powerful approach towards enabling novel functionality entails modifying oligonucleotides with chemical groups possessing intrinsic properties compatible with the particular desired function. One area with several successful examples involves incorporating photoreactive nucleobases to enable nucleic acids with optical properties. These optical properties can be particularly useful if localized DNAzyme-substrate activity can be determined and mapped. Moreover, in contrast to chemically conjugating similar chemical species to a material surface where heterogeneity in topography, local surface group distributions, etc. is typically unavoidable, the compositional uniformity in each nucleotide building block of a modified oligonucleotide sequence promises more precise spatial control of functional groups. DNAzyme activity across spatial and temporal dimensions could serve, for example, as useful “on/off” switches within imaging, sensing and therapeutic schemes. Early efforts in this regard employed photocleavable linkers in oligonucleotide backbones that effectively block the active conformation of DNAzymes. Upon irradiation by light, the linking moiety separates and enables the associated catalytic sequence to achieve its active conformation and concomitant catalytic activity. This strategy is generally referred to as photocaging and several catalytic oligonucleotides have been successfully modified with photoresponsive molecules [115,116]. By incorporating an overhang or toehold segment complementary to the catalytic region of the popular 10–23 DNAzyme, Kamiya et al. effectively inhibited its RNA cleaving ability through conformational changes induced by hybridization within the overhang segment. Photolabile azobenzene derivatives within the overhang sequence could then be converted to a *cis* conformation through light irradiation. The *cis* isomeric state destabilizes the overhang-DNAzyme duplex and liberates the catalytic domain of the DNAzyme to assume its active conformation, catalyzing RNA hydrolysis [117].

Stimuli-responsive DNAzymes might also find compelling uses for real-time monitoring of metal cations in vivo. To this end, Yang et al. constructed a photocaged Zn^2+^ specific DNAzyme by hybridizing it to a substrate strand [118]. Normally, the DNAzyme would cleave the substrate strand but this activity was blocked by modifying the substrate strand at its cleavage site, an adenosine residue, with a 2′-nitrobenzyl group at the 2′-OH position in the sugar group. The substrate strand was additionally capped with a quencher and fluorophore to initially quench the substrate strand. To arm the device after injection into zebrafish larvae, the 2′-nitrobenzyl moiety of the substrate strand was removed with UV light to deprotect the substrate strand and make it now susceptible to cleavage to release its fluorophore and thereby generate localized measurable fluorescence. Lastly, to address issues with limited light penetration into living tissues, the construct was conjugated to a lanthanide doped upconverting nanoparticle. This work demonstrates how DNAzymes modified with photolabile groups offer unique opportunities to develop spatiotemporally controlled biomolecules. Moreover, the generalization of these concepts can guide the future construction of artificial oligonucleotide devices in which specific binding events and catalytic functions are precisely dictated through exogenous stimulation.

Finally, given the original motivation to chemically modify nucleic acids decades ago, it seems appropriate to conclude with examples of the in vivo stability demonstrated by specific nucleic acid catalyst systems. Chakravarthy et al. incorporated thiophosphate and LNA into the backbones of modified DNAzymes targeting integrin-alpha-4 RNA transcripts, a target for multiple sclerosis treatment [119]. The researchers found that these DNAzymes exhibited increased nuclease stability though their performance as catalysts was reduced. Similar to findings using the enantiomer analogue of a DNA or RNA aptamer sequence, the more biostable l-RNA analogue of any D-ribozyme can be theoretically designed to recognize a target’s mirror image. Moreover, to overcome challenges posed by any resulting racemic mixtures, the Joyce group identified a cross chiral RNA polymerase ribozyme [120].

### 3.3. Future Outlook

The remarkable capabilities of nucleic acids have inspired the design and creation of a variety of functional synthetic oligonucleotides ranging from in vivo ASO and in vitro probes to non-nucleotide target-binding aptamers and completely artificial catalytic DNAzymes. While this development has established historically important oligonucleotides as viable biomolecular tools, the natural chemistry of nucleic acids is limited and can restrict the breadth of their application potential. Fortunately, the simple yet effective structure of any natural DNA oligonucleotide as a biomacromolecular template lends itself well to further expansion and experiments to explore new physiochemical possibilities. Additionally, the integration of modified functional oligonucleotides within contiguous fields of nucleic acid nanotechnology can help overcome current obstacles. For example, an emerging class of NANP, identified as promising devices for delivering therapeutics into cells, self-assemble into 3D nanostructures. Conjugating NANP with modified aptamers or DNAzymes could allow one to tailor highly functional delivery systems. Alternatively, modified oligonucleotides could be used to construct NANP to overcome immunostimulatory challenges [121,122,123].

Indeed, recent efforts to expand the chemical complexity of oligonucleotides have resulted in not only an improvement in target-binding performance and in vivo stability, but have also led to discoveries of unprecedented nucleic acid function. Recent progress in efficient polymerase engineering and template-directed synthesis methods can facilitate screening of multiple sequences in parallel through direct in vitro selection with modified libraries. Alternatively, bottlenecks in engineering polymerases that tolerate increasingly exotic chemical groups could be mitigated by employing a non-evolutionary, competition-based aptamer screening approach reported recently by the Milam group [124,125]. Notably, modern advances in ab initio modeling, machine learning and deep sequencing technologies can carve future landscapes of chemically diverse nucleic acid sequences and a rational design pathway to novel superior ligands and catalysts possessing functions unrealized by nature.

## Figures and Tables

**Figure 1 molecules-25-04659-f001:**
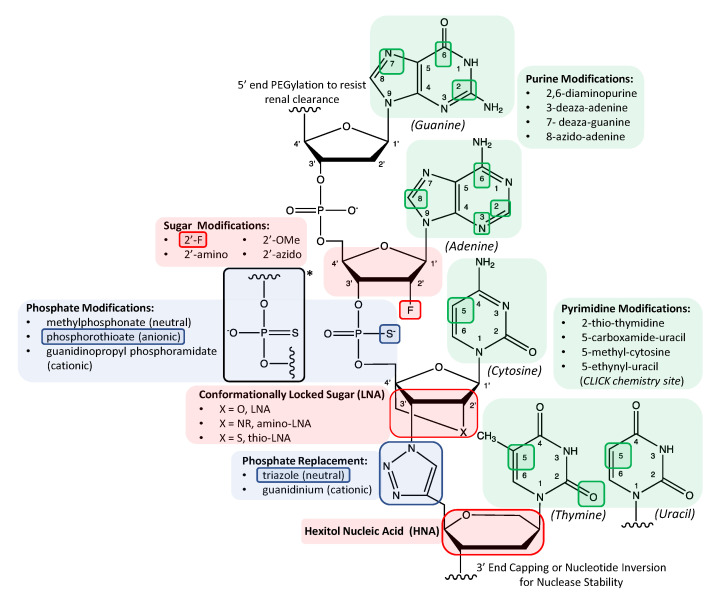
Schematic illustrating various chemical modifications to the tripartite deoxyribonucleic acid structure ranging from simple site-specific atomic substitutions to more exotic molecular replacements bearing little resemblance to the natural structure. Examples of alterations to the sugar ring component include 2′C modifications [8,9,10] (top red shaded area), conformationally locked modifications [11] (middle red shaded area) and complete substitutions of the ribofuranose ring [12] (bottom red shaded area). Other backbone modifications include atomic substitutions to the phosphate group [13] (top blue shaded area) and substitution of the phosphodiester linkage entirely [14] (bottom blue shaded area). Modifications sites in heterocyclic structures are presented for both pyrimidine and purine nucleobases (green shaded areas). * indicates alternate phosphorothioate chemical structure disputed by Frey and Sammons [15] and Liang and Allen [16]. This disputed chemical structure is still occasionally shown in recent literature including that of chemical vendors.

**Figure 2 molecules-25-04659-f002:**
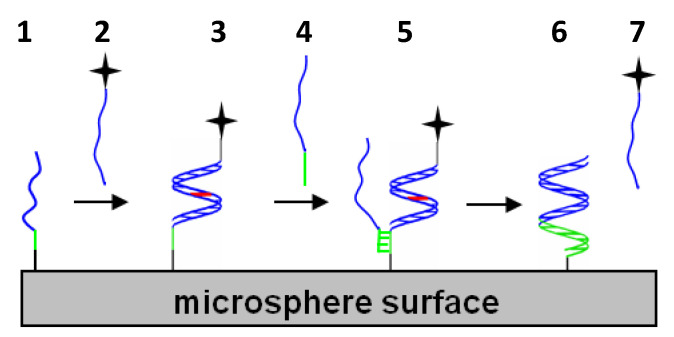
Schematic illustrating series of events for (1) microsphere-immobilized single-stranded probe incubated with (2) a labeled primary target to form (3) a labeled primary duplex with a center mismatch shown in red. Upon incubation with (4) a longer secondary or competitive target, this competitive target (5) nucleates a secondary duplex by binding to the initially unhybridized toehold segment of the immobilized probe (shown in green) and ultimately proceeds to form (6) an unlabeled, perfectly-matched secondary duplex by (7) displacing the original, shorter primary target. (Reprinted with permission from Hardin and Milam [61] Copyright 2013 American Chemical Society).

**Figure 3 molecules-25-04659-f003:**
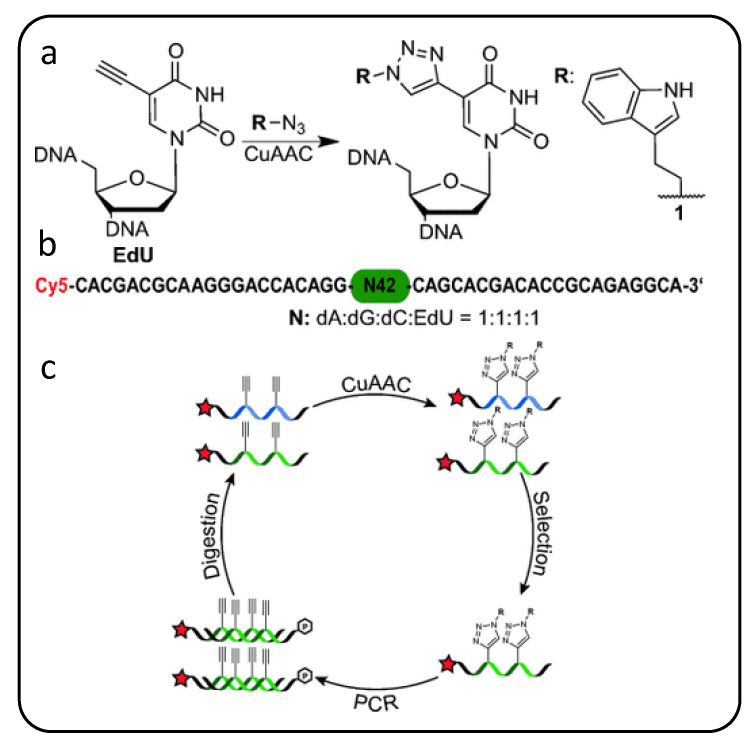
Schematic illustration of modified sequence libraries incorporating synthetic handles to side-step PCR amplification issues during click-SELEX. (**a**) To chemically modify nucleotides, C5-ethynyl handle-modified uracil nucleobases in the precursor sequences are converted to chemical groups (shown as **R** in **1**) using click chemistry to prepare (**b**) random sequence library. (**c**) During click-SELEX a modified library is incubated with a target (not shown). Following selection and recovery of target-binding sequences, the modified library must be “reverse transcribed” back to the precursor sequences bearing only the C5-ethynyl handle to facilitate PCR amplification-based enrichment of aptamer candidates. Finally, the now enriched candidate sequence population is subjected to a click reaction once more to reintroduce the chemical modification and begin the next selection cycle. (Reprinted with permission from Tolle et al. [89] Copyright 2015 WILEY-VCH Verlag GmbH & Co. KGaA, Weinheim).

**Figure 4 molecules-25-04659-f004:**
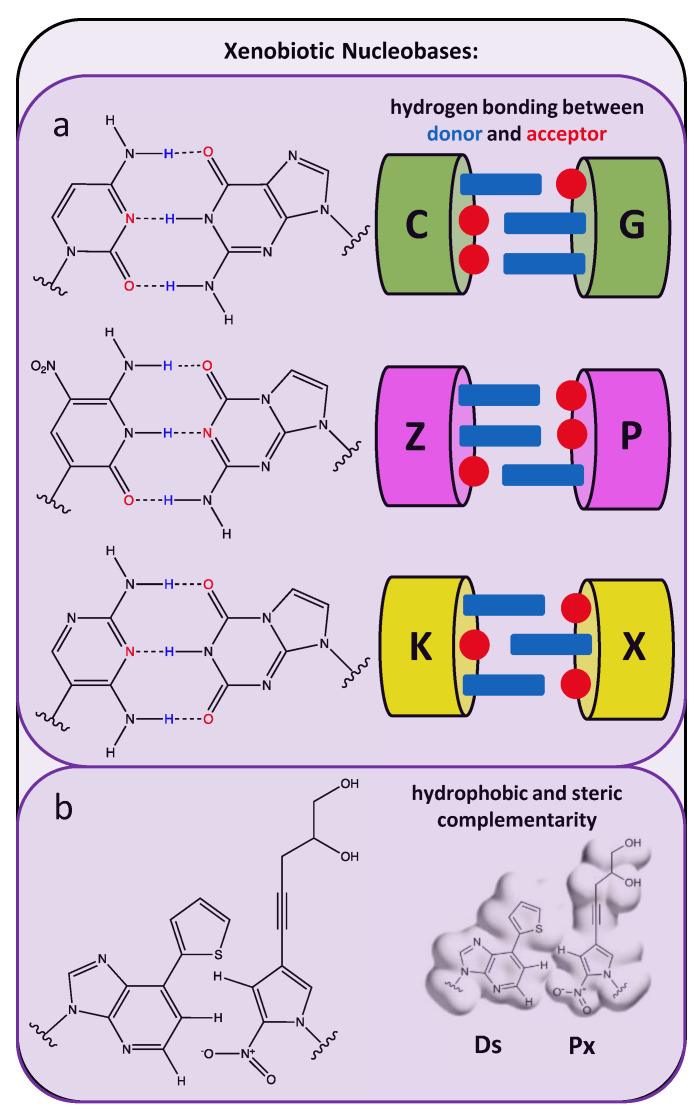
Examples of artificial nucleobase chemistries with non-Watson–Crick base-pairing complementarity. (Box (**a**)) Compared to the placement of hydrogen bond donors and acceptors notated in a natural C:G base-pair, one hydrogen bond donor and acceptor are repositioned in xenobiotic Z:P and K:X nucleobase pairs. (Reprinted with permission from Sefah et al. [96]) (Box (**b**)) Spatial alignment of hydrophobic groups in Ds and Px bases induce complementary pairing. (Reprinted with permission from Nature Publishing Group, *Nature Biotechnology*, Kimoto et al. [98] Copyright 2013).

**Figure 5 molecules-25-04659-f005:**
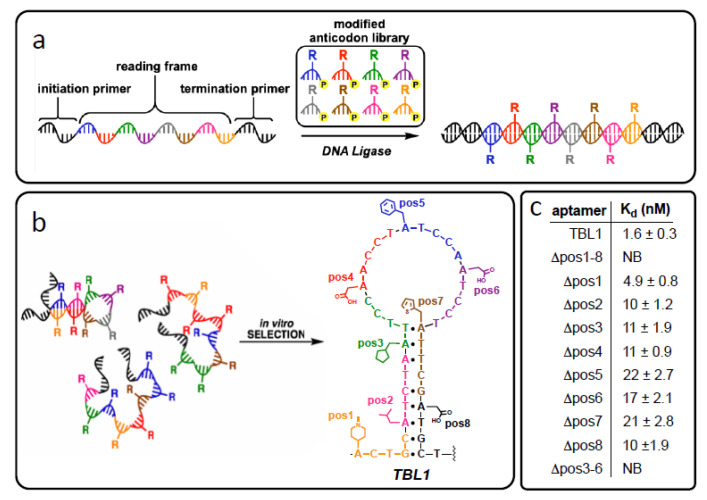
(Box (**a**)) Synthesis of densely modified oligonucleotides by ligase-catalyzed oligonucleotide polymerization (LOOPER) from pentanucleotide segments each bearing one chemical modification. (Box (**b**)) Example of a winning sequence (TBL1) isolated after in vitro selection steps with color coded anticodons and their respective modifications illustrated. (Box (**c**)) Quantitative analysis of the TBL1 LOOPER aptamer binding affinity to indicate how removing individual modifications in Box (**b**) affects the equilibrium dissociation constant, *K_d_*, to demonstrate the role of chemical modifications to TBL1 binding function. (Reprinted with permission from Kong et al. [94,103] Copyright 2020 and 2017 American Chemical Society).

**Table 1 molecules-25-04659-t001:** Examples chemical modifications and effects on various oligonucleotide properties.

Modification		Nuclease Resistance	Polymerase Compatibility	Duplex Stability	Watson–Crick Base Pairing	Ref.
Sugar	2′-F	Increase	Yes	Increase	Yes	[17,18]
	2′-OMe	Increase	Yes	Increase	Yes	[17,18]
	2′-NH_2_	Increase	Yes	Decrease	Yes	[17,18]
	LNA	Increase	Yes	Increase	Yes	[18,19]
	HNA	Increase	Yes	Increase	Yes	[12]
Phosphodiester Linkage	triazole-linked	Increase	No	Decrease	N/A	[20]
	PS	Increase	Yes	Decrease	N/A	[21,22]
	phNA	Increase	Yes	Decrease	N/A	[23]
Base	7-deaza-dA	Increase	Yes	Decrease	No	[24]
	Z/P	Unreported	Yes	Increase	Yes (modified)	[25]
	Ds/Px	Unchanged	Yes	Increase	No	[26]
	5-isobutyl-carboxamide-dU	Unreported	Yes	Increase	No	[27]

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
