# Peer review of "Modified Nucleic Acids: Expanding the Capabilities of Functional Oligonucleotides"

_molecules, 2020, doi:10.3390/molecules25204659_

Round 1
Reviewer 1 Report
The review by Ochoa and Milam nicely summarizes the recent developments in chemically modified oligonucleotides intended for biomedical applications. The manuscript is well written and will be of interest to the broad readership as it highlights and compares various therapeutic approaches, biochemical techniques, and chemical modifications including those in phosphodiester linkages, nucleobases, and sugar groups in ASOs, aptamers, and ribozymes just to name a few.
Although the authors overall did a great job reviewing the field, they didn’t mention any immunological responses that can be induced or altered due to chemical modifications. This addition would be essential for the readership as it would introduce a very critical indicator of possible clinical applications of different modifications.
The review mostly focuses on ASO and aptamer associated modification but it would be beneficial to also consider other pharmaceutically relevant examples. Among other therapeutic nucleic acids, the authors should discuss the novel class of therapeutic entities called NANPs (therapeutic nucleic acids) that represent nanoassemblies made exclusively of oligos that can perform different therapeutic functions. Adding a section about NANPs would greatly enhance the novelty of the manuscript.
Other suggestions:
- Please add a summary table with all discussed modifications highlighting their main differences, advantages, and disadvantages for use in therapeutic oligos.
- In Figure 1, please add the purpose to each modification, e.g. 5’ end PEGylation to resist renal clearance.
- Please add references for line 123-125, line 136-140, line 646-648
- The effects of 2’ modifications on RNA folding and function should be discussed as it may disrupt some non-canonical base pairings that involve sugar-edge interactions in RNAs.
- When FDA approved therapeutic nucleic acids are mentioned it would help to specify the intended mechanism of action and the target disease.
- For figure 5, box B should have the numbering introduced for all bases in order to correlate better with box c.
Reviewer 2 Report
This is a well-written review of major chemical modifications to nucleic acids. Most of the notable modification to nucleobases, sugar portion, and phosphodiester linkages are dealt with particularly in the context of antisense, aptamer, and ribozyme/DNAzyme technology. I believe it is sufficiently interesting and informative for wide readership. Some suggestions for minor revision:
- Lines 5, 6; it looks author affiliations miss some information.
- Line 538; missing closing double quotations.
- Line 581; redundant "due to" and "because of".
- Line 663; to?
- Lines 711-722; should be deleted.
- There are only subsection titles, but 2. would be more suitable for present 2.1 (and 3. for 3.1 as well).
Author Response
Please see the attached file which included responses to Reviewer #2 as well as a short list of additional text changes made to the manuscript.
